# Conversion of Glycerol to Value Added Products in a Semi-Continuous Batch Reactor Using Noble Metals Supported on ZSM-11 Zeolite

**DOI:** 10.3390/nano11020510

**Published:** 2021-02-17

**Authors:** Eliana Diguilio, María S. Renzini, Liliana B. Pierella, Marcelo E. Domine

**Affiliations:** 1Centro de Investigación y Tecnología Química (CITeQ), UE CONICET–Universidad Tecnológica Nacional, Facultad Regional Córdoba, Maestro Lopez esq Cruz Roja Argentina, Ciudad Universitaria, Córdoba 5016, Argentina; ediguilio@frc.utn.edu.ar (E.D.); mrenzini@frc.utn.edu.ar (M.S.R.); 2Instituto de Tecnología Química (UPV-CSIC), Universitat Politècnica de València, Consejo Superior de Investigaciones Científicas, Avda. de los Naranjos s/n, 46022 Valencia, Spain

**Keywords:** Au, Pt and Pd, noble metal supported zeolite, ZSM-11 microporous zeolite, glycerol oxidation, lactic acid

## Abstract

Au, Pt, and Pd supported on ZSM-11 microporous zeolite were investigated as catalysts for glycerol (GLY) oxidation towards higher value added products. ZSM-11 was synthesized by hydrothermal treatment. Subsequently, ion exchange with NH_4_Cl was performed to recover acidic sites and then, Au, Pt, and Pd were incorporated onto this material by wet impregnation procedure. After thermal treatment of desorption and calcination, the corresponding Au, Pt, and Pd-ZSM-11 catalysts were obtained. These materials were characterized by different techniques, such as XRD, ICP, TEM- XEDS, and XPS, and were evaluated in the glycerol oxidation reaction by using alkaline medium and molecular oxygen as oxidizing agent. The higher conversion of GLY (66.5 mol.%) was reached for the Pt–ZSM-11 catalyst with moderate selectivity towards lactic acid (LA), while the bimetallic Au-Pt-ZSM-11 catalyst offered high selectivity to LA at moderate GLY conversion. Optimization of the main reaction parameters (i.e., temperature, reaction time and NaOH/GLY ratio) was carried out to maximize the selectivity towards the LA desired product. Thus, LA selectivity values close to 55% at GLY conversion >65% can be reached by using Pt-ZSM-11 as a catalyst under mild reaction conditions.

## 1. Introduction

The global growth of biofuel use as biodiesel has enabled some chemicals, such as glycerol (a by-product of the biodiesel industry), to be made available in considerable quantities in the market. It is well known that for every 10 tons of biodiesel, 1 ton of crude glycerol is generated. Global glycerol production is estimated to reach approximately 3 million tons by 2020, while the total demand will not be more than 500 thousand tons [1].

The importance of the valorization of glycerol, as a platform molecule, to obtain products of interest and of commercial value has been well known in recent years. In other words, value added products can be generated, thus closing the biodiesel production cycle [2,3,4]. In this work, we focused on the valorization of glycerol (GLY) by liquid phase oxidation. This route has received a lot of attention in recent years, due to the wide variety of fine chemical products that can be produced from it. Some of these have already been reported: dihydroxyacetone (DHA), hydroxypyruvic acid (HPA), oxalic acid (OA), glyceraldehyde (GLA), glyceric acid (GA) and tartronic acid (TA), lactic acid (LA), acetic acid (AA), and formic acid (FA), among others [5,6,7].

Numerous studies have shown that both the metal, the selected support, and the operating conditions control the conversion and selectivity to the reaction products. Extensive homogenous and heterogeneous catalysts have been designed and evaluated in liquid phase GLY oxidation.

In recent decades, progress has been made in the search for higher yields towards desired products through the design of suitable catalysts, allowing the proposal of a possible reaction mechanism and network [8,9,10]. Among different reaction systems, operating conditions and different solid catalysts, noble metals (Au, Pt and Pd) supported on various solid catalysts are mostly reported. On one hand, catalysts based on activated carbon [11,12,13] achieve the best results, with GLY conversion of ≈99% with selectivity towards LA of 68% and 74% when using 10%Pd/C and 5%Pt/C as catalysts, respectively, by working at 230 °C in a stainless-steel Parr reactor. In addition, Pd and Pt supported on graphite and carbon nanotubes reached a GLY conversion of 90% and GA selectivity of about 60–70% after 5 h of reaction [14,15]. On the other hand, Au-Pt bimetallic catalysts were supported on different supports, including acidic (TiO_2_ and CeO_2_), basic (MgO, Mg(OH)_2_, (MgCO_3_)_4_, Mg(OH)_2_, ZnO, and CaCO_3_), and amphoteric supports (Al_2_O_3_ and hydrotalcite) with the purpose to study the effect of acid/base properties of the supports on the performance of catalysts in glycerol oxidation. The results showed that the properties considerably affected the selectivity of products [16,17]. Double lamellar hydroxides [18] and other supports such as zeolite-type materials have also been evaluated [19]. Therefore, the choice of a suitable support is essential to achieve the objective in terms of obtaining the desired products.

It is well-known that heterogeneous catalysts offer several advantages compared to homogeneous catalysts, such as easy separation of oxidation products, possibility of recovery and regeneration, more environmentally friendly systems, and minimization of purification steps, among others. Nevertheless, the effectiveness of the heterogeneous catalyst must be equivalent to that of the homogeneous catalyst. A key parameter here is the number of reaction cycles per active site that the catalyst is able to make, as well as the stability of the catalyst under reaction conditions. 

It is important to mention that very few studies in the literature have identified noble metal supported zeolites used as catalysts for the selective oxidation of GLY in liquid phase. On the other hand, we previously evaluated zeolites such as ZSM-5, ZSM-11, BETA, and Y with the incorporation of Cu and Cr transition metals in GLY oxidation under mild conditions. When Cu-ZSM-11 was used as a catalyst, GLY conversion close to 30% and 68% of LA selectivity was reached after 1 h of reaction by using H_2_O_2_ as an oxidant, whereas ≈40% of GLY conversion and 30% DHA selectivity were achieved after 2 h of reaction with Cr-ZSM 11 as catalyst [19].

Based on these findings, it is of great interest to design a catalytic system using Au, Pt, and Pd supported on ZSM-11 type zeolites capable to oxidize GLY selectively. A great challenge is to obtain a less-expensive process (including both catalyst preparation and GLY oxidation) than those previously described. In this sense, the use of less-expensive oxidants, such as molecular O_2_ or air, as well as the employment of mild reaction conditions (low temperature and pressure) are highly preferred. In the case of a catalyst, the use of zeolite ZSM-11 (MEL) as support is proposed, which is an important member of the pentasil zeolite family—similar framework density and pore size with another family member, ZSM-5. The zeolites in this family are highly attractive due to their unique catalytic and adsorbent properties, the MEL material hardly being reported in the literature in relation to zeolites with MFI structure. Zeolite ZSM-11 is a solid material with shape selectivity properties due to its structure and molecular size of its pores, together with high chemical and thermal stability. In addition, the presence of Brønsted and Lewis acid sites provides interesting acid properties to the material, these properties being relatively ease to modify by incorporating active functions through exchange/impregnation with different metal cations. On the other hand, the incorporation of lower percentages of metal could be an advantage if, in turn, the catalyst is active and selective to the desired product, as well as highly resistant to reuse. In addition, the use of mild reaction conditions and a non-toxic less-expensive oxidant (i.e., molecular O_2_ or air) during the process will be other goals to achieve, thus facilitating the industrial implementation of the process.

Among the different oxidation products that can be obtained from GLY, this study will focus on lactic acid, 2-hydroxypropionic acid, which is an organic acid widely found in nature. Lactic acid (LA) is used in the food and chemical industries, as well as in pharmaceuticals, cosmetics, textiles, and as a monomer to obtain the biodegradable polymer polylactic acid [20,21,22]. LA can be obtained by chemical or fermentative routes, the latter being chosen at the industrial level, since high yields are obtained at low reaction temperatures, but it needs to operate in dilute conditions to avoid excessively low pH conditions. The purification process also produces large amounts of waste without commercial interest. Therefore, the development of new routes for the production of lactic acid has aroused great interest within the research community, aiming at a reduction in production costs and environmental impact [23,24].

The development and optimization of adequate recovery and purification systems are key factors influencing effectiveness in lactic acid production, since purification represents a large percentage of the total cost. Despite it being considered as a very interesting alternative, there are no examples of industrial processes based on GLY as a starting material to obtain LA. For this reason, it is highly interesting to develop a heterogeneous catalyst that is active and selective towards the desired product, as well as having great resistance and reuse capacity.

This study provides insights into the development of metal-supported catalysts based on the combination of well-dispersed nanoparticles of noble metals, Au, Pt, and Pd adequately incorporated into ZSM-11 type zeolite synthesized materials used for the first time as catalysts for application in the selective oxidation of GLY towards lactic acid (LA). Particularly, reactions are performed under very mild reaction conditions (at low temperature and atmospheric pressure) and by using molecular O_2_ as an oxidant, thus offering the possibility to obtain an interesting product, such as LA, at industrially applicable conditions.

## 2. Materials and Methods

### 2.1. Synthesis of Catalytic Materials

ZSM-11 zeolite (Si/Al molar ratio of 17) was synthetized in a stainless-steel reactor by hydrothermal treatment at 140 °C, using sodium aluminate (NaAlO_2_) and silica (SiO_2_) as sources of aluminum and silicon, respectively, with an aqueous solution of tetrabutylammonium hydroxide (TBAOH 30% *v*/*v*) as the structure-directing agent [25]. The molar composition of the resulting gel was: SiO_2_ = 33.90; Al_2_O_3_ = 1.00; Na_2_O = 1.25 (TBA)_2_O = 3.20 and H_2_O = 700. The gel from the synthesis was separated by vacuum filtration, washed with deionized water up to neutral pH, and then dried at 100 °C overnight. To remove organics compounds, a thermal treatment was carried out, first a desorption under N_2_ flow (20 mL/min) from 25 °C up to 500 °C, for 8 h. Then, the catalyst was calcined at 500 °C for 12 h. Thereby, the Na-ZSM-11 catalyst was obtained. To obtain the NH_4_-Zeolite form, Na-ZSM-11 was ion exchanged with ammonium chloride aqueous solution (NH_4_Cl, 1 M) at 80 °C for 40 h. Finally, NH_4_-ZSM-11 was impregnated with precursor salts of Au, Pt, and Pd.

### 2.2. Incorporation of Noble Metals

The synthetized zeolites were modified by the incorporation of ~1 wt.% of noble metals such as Au (III), Pt(VI), and Pd(II). The precursor salts were HAuCl_4_·3H_2_O (99.9%, Tetrahedron), Pd(CH_3_COO)_2_ (49%, Sigma Aldrich, Sant Louis, USA), and PtCl_4_ (96%, Sigma Aldrich, Sant Louis, USA), respectively. They were incorporated in NH_4_-ZSM-11 by the wet impregnation method. The procedure consisted of contacting an aqueous solution of the corresponding metal precursor with the zeolite in ammonium form (NH_4_-ZSM-11). For bimetallic systems, the co-impregnation method was employed. Then, the samples were evaporated in a rotary evaporator at 80 °C and 30 rpm until complete dryness. Finally, the catalysts were dried at 100 °C overnight, and then exposed to thermal desorption under N_2_ atmosphere and calcination treatment at 500 °C for 8 h, to remove organic residues of precursor salts.

### 2.3. Characterization of the Catalysts

The synthetized catalysts were characterized by different techniques, to identify their physical and chemical properties. First, X-ray diffraction (XRD) measurements were used to determine the structure and crystallinity of the materials employing a Philips PW 3020 diffractometer using Cu Ka radiation. Diffraction data were recorded in the 2ϴ range 2–60° at an interval of 0.11 and scanning speed of 2°/min was used.

For the structural characterization in vibration zone (400–1800 cm^−1^), infrared spectroscopic analysis was performed using a JASCO 5300 FTIR spectrometer. To determine the type and concentration of the acidic sites, pyridine (probe molecule) adsorption experiments were carried out on self-supporting zeolite wafers (10–20 mg/cm ^2^) using a thermostatic cell with CaF_2_ windows connected to a vacuum line. Pyridine (3 Torr) was adsorbed at room temperature and desorbed at 250, 350, and 400 °C for 1 h. Finally, the Brønsted (B) and Lewis (L) acid sites was calculated from the adsorption bands with maximum intensity (1545–1555 and 1450–1460 cm^−1^, respectively). The B/L ratio was calculated by using the literature data of the integrated molar extinction coefficients, which are independent of the catalysts or strength of the sites [26].

Surface area of the catalysts was calculated from the N_2_ adsorption isotherms by using the BET method in an ASAP 2000 equipment with N_2_ absorption at 77 K. The metal content in ZSM-11 samples was determined by using an inductively coupled plasma-atomic emission spectrometer, ICP Varian 715ES.

To determine the size and dispersion of metal particles on the zeolite surface, transmission electron microscopic (TEM) measurements were performed in a JEM 2010 Plus electron microscope, equipped with an EDS, Oxford X-MAX 65 T, operating at 200 kV. The samples were pre-treated by ultrasonic dispersion in ethanol and then dropped onto grids.

X-ray photo electronic spectroscopy (XPS) analysis of the synthetized catalysts were performed to determine the concentration and type of metallic species present on the surface of the catalysts. XPS spectra have been obtained with a Thermo Fischer Scientific Model K alpha kit, equipped with an Al (Kα) anode source operating at 1200 eV. An energy step of 40 eV to 0.1 eV was used. The spot size was 400 μm. The binding energy of the C 1s peak (BE 284.6 eV) was used as reference. 

### 2.4. Catalytic Test

The selective oxidation of GLY (98.8% *v*/*v* Merck, ACS reagent) in the liquid phase was evaluated employing mono and bimetallic catalysts, such as Au, Pt, and Pd supported on ZSM-11 zeolite. The reaction was carried out in a glass reactor (50 mL) with three necks connected to a condenser and a magnetic stirrer. The system was immersed in a water/silicon bath to homogenize the temperature inside the reactor throughout the experiment. 

An aqueous solution of GLY (35 mL, 0.25 M) was mixed in the reactor with NaOH (NaOH/GLY = 2 mol/mol) and 200 mg of the catalyst (GLY/metal = 400 g/g). After reaching the desired temperature (70 °C), the oxygen bubbling (100 mL/min) starts within the solution using a glass frit. Reactions were done under continuous stirring at optimized stirring rate (>800 rpm) in order to avoid mass transfer limitations. 

The reaction products, as aldehydes, ketones, and acids were identified and quantified by means of high-pressure liquid chromatography (HPLC) analysis of liquid samples with a Jasco UV-975/PU-980 equipment. Two detectors coupled in series: UV-Vis detector (210 nm) and refractive index (RI—40 °C) were used. The column (Aminex HPX-87H) was set at 50 °C. An aqueous solution of sulfuric acid (5 mM) was employed as mobile phase (0.6 mL/min). The quantification of the oxidation products was done by calibration curves of pure standards.

Mass balances were calculated in all the reactions, mostly yielding values >95. Minimum mass loss could be attributed to solvent evaporation and sample handling during the experiment. In the cases of mass balance being lower than 95%, the experiment was discarded.

The reaction conditions’ optimization was carried out by employing 35 mL of aqueous GLY solution 0.25 M and 200 mg of Pt-ZSM-11 as catalyst, with the aim to obtain maximum selectivity toward the desired product. The reaction temperature ranged from 70 to 100 °C, NaOH/GLY molar ratio was varied from 1 to 4, and reaction times in the range from 1 to 6 h, in all catalytic tests.

## 3. Results and Discussion

### 3.1. Characterization of the Catalysts

The main physico-chemical and textural properties of noble metal supported zeolites prepared in this study are shown in Table 1. The surface areas of the solid materials were calculated from N_2_ adsorption isotherms by using the BET method, while the metal contents (wt.%) of the mono and bimetallic catalysts were determined by ICP technique. The type and amount of acid sites present in the catalysts were measured by FTIR with pyridine adsorbed at room temperature and then desorbed at 400 °C (see Experimental section).

As shown in Table 1, the surface area values of the catalysts were close to that of the starting matrix (ZSM-11), with slight decrease in some cases. This could be because the metal particles are deposited on the surface of the zeolite crystals, thereby decreasing the surface area eventually [27] (pp. 321–322).

The measured metal contents resulting in metal incorporation, either by impregnation or co-impregnation, were found to be close to the theoretical value calculated for all catalysts (1 wt.%). In addition, the relationship of Brønsted/Lewis acid sites of all catalysts are reported in Table 1. As can be seen, the Brønsted/Lewis ratio decreased with noble metal incorporation onto the pristine ZSM-11 zeolite; this decrease was more evident in the case of bimetallic systems.

Figure 1 shows the X-ray diffraction patterns of the ZSM-11 matrix and the same after incorporation of noble metals Au, Pt, and Pd by means of wet impregnation. It can be seen that all diffractograms are those corresponding to highly crystalline MEL-type zeolitic structures, with well-defined signals of high structural order (2θ = 7–8° and 2θ = 23–24°). No changes in the crystalline structure were observed after incorporation of the metal species, but in Au-ZSM-11 diffractogram, the presence of signals corresponding to Au^0^ species (*) incorporated was clearly observed. However, in Pt-ZSM-11 and Pd-ZSM-11, signals corresponding to Pt^0^ (+) and Pd^+2^ (°) were observed, respectively.

The diffractograms of the bimetallic zeolites Au-Pt-ZSM-11 and Au-Pd-ZSM-11 are depicted in Figure 2, where the signals corresponding to Au^0^, attributed to the crystallographic plane (111) at 2Ɵ = 38.18°, and (200) at 2Ɵ = 44.38°, can be observed. In addition, the signal corresponding to species Pt^0^ (+), attributed to the plane (111) at 2Ɵ = 39.7° and (200) at 2Ɵ = 24.23°, can be distinguished in the Au-Pt-based sample.

For the Au-Pd-ZSM-11 bimetallic catalyst (Figure 2), the characteristic peaks assigned to PdO (Pd^+2^) (°) corresponding to plane (101) at 2Ɵ = 33.9° and (112) at 2Ɵ = 54.8° were observed. All the signals have been indexed with the characteristic diffraction pattern using Highscore Plus software, PANalytical, The Netherlands (XRD Analysis Software [28].

Furthermore, the morphology, distribution, and particle size of mono and bimetallic catalysts were determined by transmission electron microscopic (TEM) measurement and energy dispersive X-ray spectroscopy (XEDS), to obtain a representative metal particles composition on the catalytic surface (see Appendix A). TEM images of the samples are presented in Figure 3 and Figure 4, where the dark spots correspond to the metal particles that have been incorporated in the ZSM-11 zeolite in each case.

With the aim to investigate the dispersion and average size of metal particles, TEM micrographs were analyzed by using the ImageJ NIH program, to obtain particle size distribution considering more than fifty nanoparticles in each case.

TEM images of the monometallic catalysts (Figure 3) show the presence of metal particles on the ZSM-11 surface, these results being in line with those obtained by XRD measurements. For the Au-ZSM-11 catalyst, TEM images show a narrow particle size distribution with an average diameter of 3 nm. Low-scale images (20 nm) confirm the presence of well-dispersed nanoparticles on the surface, specifically, a large number of small particles with spherical morphology [29,30].

For Pt- and Pd-based catalysts, metal particles had an average diameter of 16 and 22 nm, respectively. The biggest metal particles (28 to 38 nm in diameter) were also found. As can be seen, the formation of agglomerates on the zeolite crystals (darker spots) could also be observed, mainly in the Pd-ZSM-11 catalyst (Figure 3c) [10]. 

On the other hand, TEM images, size distribution of metal particles, and XEDS analysis of bimetallic Au-Pt- and Au-Pd-ZSM-11 catalysts are presented in Figure 4 and in Appendix A.

For both catalysts, a particle size distribution is observed between 1 to 10 nm, with an average diameter of 3.4 nm for Au-Pt- and 4.4 nm for Au-Pd-ZSM-11 catalysts, respectively. TEM images of both catalysts shows darker wide areas, which would be indicative of the interaction between the two metals (Au-Pt and Au-Pd). By XEDS analysis of a representative area on the catalytic surface, the closeness of metallic species Au, Pt, and Pd could be observed, especially in bimetallic catalysts. This could indicate the formation of alloys of Au-Pt and Au-Pd. A more detailed study should be performed to confirm the presence of such an alloy by net spacing calculation [8,31,32].

In addition, X-ray photoelectron spectroscopy (XPS) was used to study the surface chemical composition of the catalyst and the oxidation state of the noble metals. The XPS spectra of mono and bimetallic catalyst are presented in Figure 5 and Figure 6. In each case, C 1s core level (284.6 eV) was used for calibration of the energy scale as reference. Al 2p (74.4 eV) and Si 2p (100–110 eV), both present in the zeolite matrix, were also detected. 

XPS spectra of the Au-ZSM-11 monometallic catalyst shows two doublets for the Au 4f, corresponding to the Au^0^ (84.4 eV) and Au^+1^ (86.7 eV) species, with the major contribution corresponding to Au^+1^ [29,33].

The presence of Pt^0^ in the Pt-ZSM-11 catalyst was evidenced by the contribution of Pt 4f_7/2_ at 71.5 eV with an atomic percentage of 0.1% [30]. Meanwhile, in the spectra corresponding to Pd-ZSM-11, the sample Pd 3d_5/2_ was composed by two contributions at 336.9 eV and 337.9 eV assigned to Pd^0^ and Pd^+2^, respectively. The atomic percentage was around 0.1% in both species [27]. These results were also corroborated by programmed temperature reduction (TPR), where the absence of a signal in the Au-ZSM-11 and Pt-ZSM-11 catalysts would indicate the presence of the species Au^0^ and Pt^0^, respectively, and the appearance of a signal between 300–400 °C corresponds to PdO in the Pd-ZSM- 11 (see Appendix A).

On the other hand, the presence of Au in metallic state and the Pt^+2^ (72.6 eV) species with a higher atomic percentage in the surface of the zeolite [31] was observed by analyzing the spectra of the Au-Pt-ZSM-11 bimetallic catalyst. For the Au-Pd-ZSM-11 bimetallic sample, the presence of Au was evidenced by the signal Au 4f_7/2_ at 84.4 eV (Au^0^). In addition, the presence of the Pd species was corroborated by the signal mentioned above at 336.9 eV assigned to Pd^0^ and at 337.9 eV assigned to Pd^+2^, corresponding to PdO [33,34]. All these observations were also confirmed by TPR analysis (not presented here).

In summary, the surface atomic percentages of the Au, Pt, and Pd species in the catalysts determined by XPS analysis are presented in Table 2.

### 3.2. Catalytic Activity

The catalytic performances of the synthetized catalysts were investigated in the selective oxidation of GLY in the liquid phase and alkaline medium, with oxygen as an oxidant under mild conditions and without using an organic solvent. In these experiments, mono (Au, Pt and Pd-ZSM-11) and bimetallic catalysts (Au-Pt- and Au-Pd-ZSM-11) were employed, and the effect of operational reaction conditions were analyzed. The initial reaction parameters employed were aqueous GLY solution (0.25 M), at 70 °C, with NaOH/GLY molar ratio = 2, O_2_ flow of 100 mL/min, amount of catalyst 200 mg and reaction time of 4 h.

It should be noted that by using other zeolites such as H-ZSM-5, H-BETA, and H-Y, GLY conversion reached values between 15 and 25 mol.% under the same conditions. Meanwhile, the H-ZSM-11 zeolite achieved the best results in terms of GLY conversion (>65 mol.%) and greater selectivity to the desired products, even when comparing with previous results attained with other metals (i.e., Cu and Cr) supported on this zeolite [19]. Due to this, the ZSM-11 matrix was selected for this study.

Notably, the presence of a basic medium is essential for the progress of the reaction towards the oxidation products. The results obtained in a base-free medium were not satisfactory, achieving GLY conversion values between 2 to 7 mol.%. For this reason, all catalytic experiments were carried out in alkaline medium. 

Catalytic activity and product distribution obtained in GLY selective oxidation for different metal-supported catalysts are summarized in Table 3. Among the identified reaction products are glyceraldehyde (GLA), glycolic acid (GLIC), glyceric acid (GA), lactic acid (LA), acetic acid (AA), oxalic acid (OA), and in a small proportion those not identified (NI).

It can be seen that Pt-ZSM-11 achieved the highest GLY conversion (66.5 mol.%) followed by the Au-ZSM-11 (50.5 mol.%) and Pd-ZSM-11 (38.0 mol.%) catalysts. The former was shown to be selective towards LA, while GA was the major product observed with Pd-ZSM-11 and Au-ZSM-11. This behavior could be due to the higher Brønsted/Lewis relationship in Pt-ZSM-11 (B/L = 4.76), followed by Au-ZSM-11 (B/L = 4.12), and then Pd-ZSM-11 (B/L = 3.95). In addition, metal particle size on the zeolite could influence this behavior (see Figure 3 and Figure 4); small particle size results in a better dispersion of the metallic species on the catalyst surface and this causes a higher catalytic performance [35].

As mentioned, the Pt-ZSM-11 catalyst was selective in the oxidation of GLY toward LA, while the Pd-ZSM-11 catalyst resulted in higher selectivity to GA, similar to Au-ZSM-11. These results are in concordance with those reported in the literature, since both Pd and Au are more selective towards GA under basic conditions, while Pt is more selective to attain DHA and LA under mild conditions [2,36].

According to reaction mechanisms proposed in the next section, the oxidation of GLY initially involves oxidative dehydrogenation of GLY towards GLA and DHA. In alkaline medium, GLA and DHA are in equilibrium and can undergo rearrangement to LA. On the other hand, a catalyst with a high oxidizing power could cause the conversion of GLA into GA and other subsequent oxidation products [8].

Another interesting point is the combined effect between the noble metals, which indicates if a synergistic effect occurs between them. In this way, the catalytic activity of the bimetallic Au-Pt-ZSM-11 and Au-Pd-ZSM-11 catalysts was also evaluated. As can be seen in Table 3, although the GLY conversion attained with bimetallic systems was lower than that obtained with monometallic catalysts, the Au-Pt ZSM-11 catalyst achieved a ≈45 mol.% selectivity to LA, higher than that achieved with the Pt-ZSM-11 sample (LA select. = 30.4 mol.%). This behavior could be related to the lower relationship of Brønsted/Lewis acid sites present in the bimetallic samples; these values being between 29% in the Au-Pt-ZSM-11 zeolite and 44% in the Au-Pd-ZSM-11 zeolite lower than these encountered for the Pt-ZSM-11 and Pd-ZSM-11, respectively. It should be noted that LA had not been detected when using the Au-ZSM-11 catalyst. In other words, the presence of both Au and Pt in the same catalyst leads to a combined effect, directing the reaction towards obtaining LA, via DHA.

Interestingly, the Au-Pd-ZSM-11 catalyst achieved 26.9 mol.% of LA selectivity during GLY oxidation. This was not detected when the monometallic Au-ZSM-11 and Pd-ZSM-11 catalysts were used. Based on these results, it can be assumed that there is a synergistic effect between these metals, meaning that Au would act as a promoter of Pt and Pd, thus producing higher yields to LA and favoring the dehydration reaction of GLA/DHA, instead of the oxidation towards GA. This catalytic behavior could be due to the formation of an alloy between Au-Pt and Au-Pd on the ZSM-11, as mentioned above. Local composition of individual metal particles showed a close interaction between them, evidenced by XEDS spectra taken for the representative region on the crystal surface of bimetallic systems (see Appendix A) [8]. 

Comparison of the results obtained with Pt/ZSM-11 and Au-Pt/ZSM-11 with those previously reported by us with transition metals (Cr, Cu) supported zeolites. It can be said that with both reaction systems, similar yields were achieved for the desired products (see tables of S4 section in Appendix A). Particularly, LA yields between 15–20% at 1–2 h of reactions are achieved with both noble metals and transition metals supported on the ZSM-11 zeolite. Nevertheless, it is important to remark that when using noble metals supported on ZSM-11, milder reaction conditions have been selected (atmospheric pressure and low temperatures) compared to noble metals supported systems previously reported by other authors [37,38,39,40,41]. It is clear that higher reaction pressures and temperatures have an effect on the final GLY conversion and the selectivity of the products, thus overcoming the susceptibility of noble metals to poisoning and leaching in the reaction medium (see section S5 in Appendix A). However, the results reached in this work were promising and offer potential to continue finding optimal conditions to obtain higher yields.

Considering the catalytic results above discussed, the monometallic catalyst Pt-ZSM-11 was selected to study the influence of the main reaction parameters on the GLY oxidation. The temperature, reaction time, and NaOH/GLY molar ratio, as well as the reaction network were evaluated; the results are presented in the following section.

### 3.3. Reaction Pathways

The reaction network proposed in accordance with oxidation products distribution observed and the literature data is presented in Figure 7. The first reaction step consists of oxidative dehydrogenation of GLY with formation of GLA/DHA, primary oxidation products, which remain in equilibrium in basic medium. These trioses can be rearranged and later transformed into lactic acid (LA) [42]. The use of acidic solid supports, such as zeolites, modified by incorporating mono or bimetallic cations, present a Lewis acidity higher than, for example, metal oxides, thus requiring milder reaction conditions to complete acid-driven reactions such as DHA dehydration. The form selectivity of these materials as well as the presence of Lewis acid sites favor the direct conversion of trioses to lactic acid [37].

Kumar et al. [8] proposed GLA/DHA as intermediaries in the conversion of GLY to LA, followed by an undesirable (parallel) reaction involving the catalytic oxidation to GA, and later on to smaller molecules such as TA, GLIC, OA, AA, and FA, through consecutive oxidative C–C bonds splitting.

Taking this into consideration, an optimal catalyst should possess a strong dehydrogenation capacity and, at the same time, be less efficient in reactions that involve rupture of C–C bonds, thus avoiding the generation of undesirable over-oxidation products [8,23]. In addition, the presence of a base is essential in the conversion of GLY, since OH^−^ intervenes in the determining step of the reaction: the dehydrogenation of the alcohol group of GLY. These OH– groups then react with the alcohol to give the corresponding acid. Meanwhile, the molecular O_2_ (oxidizing agent) participates in catalytic cycle regenerating OH– ions necessary for oxidation [43].

In order to gain more insights into the reaction mechanism and different pathways to obtain oxidation products during the GLY oxidation process, a catalytic experiment was carried out under the reaction conditions mentioned above by using an aqueous solution of GLY + 10 wt% DHA at the beginning of the reaction. Au-Pt-ZSM-11 was selected as catalyst for this test, due to its highest selectivity to LA. The results showed that LA was detected in the early stages of reaction; the presence of GLA was also observed, but without detecting DHA. On the other hand, products such as GA, GLIC, AA, and OA were also detected, but in very low concentrations. These observations suggest that the presence of DHA in reaction medium favors LA production, via GLA/DHA dehydration and posterior molecular rearrangement, while oxidation to products with lower carbon numbers was disfavored. These results were in agreement with those reported in the bibliography [27,40].

With all these data and observations in mind, it is possible to stress that the presence of highly and homogeneously dispersed noble metals nanoparticles in our Pt/ZSM-11 and Au-Pt/ZSM-11 catalysts allow to produce the dehydrogenation reaction of GLY to GLA/DHA, helped by the OH^−^ species given by the basic medium. Then, the equilibrium GLA/DHA is shifted towards the GLA for the later production of LA via dehydration, mainly carried out onto the Brönsted/Lewis acid sites of the ZSM-11 zeolite. It is true that once the carbonyl (aldehyde or ketone) intermediate is formed on the catalyst surface, it undergoes the elimination of β-hydride to generate the corresponding carboxylic acid, which is also facilitated by the existing surface bound hydroxide formed during the first dehydrogenation step. Nevertheless, the use of very mild reaction conditions reduces to some extent in the production of other oxidation products, also avoiding the C–C rupture occurring due to over-oxidation reactions that derive in the formation of C_1_–C_2_ carboxylic acids.

### 3.4. Study of the Reaction Parameters of the Catalytic System

#### 3.4.1. Effect of Reaction Temperature

The effect of the reaction temperature in the GLY oxidation on the Pt-ZSM-11 catalyst was evaluated by varying it between 70 and 100 °C and keeping other reaction parameters constant, such as GLY aqueous solution (0.25 M), NaOH/GLY molar ratio of 2, oxygen flow rate (100 mL/min), amount of catalyst (200 mg), and reaction time (4 h). The results obtained are presented in Figure 8.

It can be seen that GLY conversion reaches a maximum of around 60–65 mol.%, after 4 h of reaction. This behavior could be due to the formation of oxidation products, which are more reactive than GLY molecule, causing an apparent loss of activity (in terms of GLY conversion) with increasing temperature [44]. 

Concerning the distribution of reaction products, on one hand, there was an increase in selectivity to LA from 30.5 mol.% (at 70 °C) to 53.5 mol.% (at 100 °C) with the increase of reaction temperature. At the same time, the selectivity to glyceric acid (GA), acetic acid (AA), and oxalic acid (OA), by-products obtained by over-oxidation pathways, decreased with the increase in reaction temperature, thus implying that the dehydration reaction rate (DHA/GLA to LA) has a stronger temperature dependence than that of the oxidation reaction rate (GLA to GA) [13,45].

With these results in mind and in order to continue with the evaluation of reaction parameters, 100 °C was selected as the optimal reaction temperature, since a high yield to the desired product LA and low selectivity to over oxidation products were obtained.

#### 3.4.2. Effect of NaOH/GLY Molar Ratio

Another operating parameter studied was NaOH/GLY molar ratio. Importantly, the presence of a base in reaction medium is essential for the progress of the reaction, since the OH^−^ groups are involved in the determining step of reaction: the dehydrogenation. The peroxide radicals (OOH * and H_2_O_2_ *) are responsible for C–C splitting, favoring the formation of by-products as OA, AA, and FA, among others. As is well known, oxygen activation occurs by the formation and dissociation of peroxide intermediates (OOH *; H_2_O_2_*) on the metallic surface of the catalyst [39]. 

Figure 9 shows that GLY conversion reached a maximum at NaOH/GLY molar ratio of 2. Thus, GLY conversion notably increased when the NaOH/GLY molar ratio varied from 1 to 2, and then, it suffered a decrease when NaOH/GLY changed from 2 to 4.

On one hand, a higher concentration of base in reaction medium favors the formation of peroxide radicals, needed for progression of the reaction. On the other hand, by increasing the NaOH/GLY molar ratio from 2 to 4, the decrease in GLY conversion could be explained by the formation of sodium Glycerolate. This take place by the combination of Na^+^ (from the base) and the deprotonated form of GLY; the formed glycerolate is less reactive than the GLY molecule. Therefore, a loss of activity, and a decrease in final GLY conversion could arise [16].

Regarding LA selectivity, it remained practically constant until NaOH/GLY molar ratio of 2, and then it decreased with an increase in the NaOH/GLY ratio from 2 to 4. This behavior is attributed to the fact that a higher concentration of base favors reaction involving degradation of LA and GA, generating C–C scission products as AA, OA, and FA [8,13,46].

From these results, it can be concluded that with a NaOH/GLY molar ratio of 2, maximum GLY conversion and LA yield were obtained, and the formation of by-products was reduced. Owing to this, NaOH/GLY molar ratio of 2 was selected as optimum value for the catalytic oxidation of GLY with Pt-ZSM-11.

#### 3.4.3. Effect of Reaction Time

Figure 10 shows the GLY conversion and selectivity towards oxidation products evaluated in function of reaction time in the GLY oxidation of the Pt-ZSM-11 catalyst. The experiments were carried out at different reaction times (for 3, 4, 5, and 6 h) without modifying the other operating parameters (100 °C, NaOH/GLY = 2, catalyst = 200 mg).

As can be seen, the GLY conversion reached its maximum after four hours of reaction (240 min), then remained practically constant at longer reaction times. The selectivity to the LA product also reached a maximum (53.5 mol.%) after 240 min of reaction, but then it sharply decreased at longer reaction times (up to approximately 20 mol.% after 360 min of reaction). This result was in agreement with the increase in selectivity to AA and OA, formed by the over oxidation of LA and GA at extended reaction times [10].

In summary, during GLY oxidation with the Pt-ZSM-11 catalyst, it was observed that GLY conversion remained invariable at extended reaction times (after 240 min), while the selectivity to LA product suffered a great decrease (from ~54 to 20 mol.%). This decay in LA yield could be mainly due to the occurrence of undesired reactions, for example, splitting of C–C bonds, with the subsequent appearance of over-oxidation products with lower molecular weight (AA, OA). For this reason, a reaction time of 4 h (240 min) was selected as the optimum value for this catalytic process. Under these conditions, LA yield was maximized and no significant amounts of over-oxidation products were detected (AA and OA selectivity lower than 5 mol.%).

### 3.5. Homogeneous vs. Heterogeneous Reaction Test

The occurrence of homogeneous vs heterogeneous reactions during the GLY oxidation on the Pt-ZSM-11 catalyst was studied by evaluating the possible loss of Pt species from the solid catalyst, by leaching within a liquid reaction solution. For that purpose, an experiment was carried out under the optimal operational conditions established in previous sections.

The catalytic test consisted of starting the GLY oxidation and then removing the solid catalyst from the reaction mixture after 60 min of reaction, by vacuum filtration. Afterwards, the GLY oxidation continued without a catalyst for 4 h, under the above-mentioned conditions. Liquid samples were taken from the reaction at different time intervals, and then analyzed by HPLC. These experimental results are presented below in Figure 11.

As can be seen, GLY conversion reached 26.3 mol.% at 60 min of reaction, and after solid catalyst removal from the reaction medium at this time, GLY conversion remained constant until the end of the reaction (240 min). At the same time, LA selectivity reached 31.0 mol.% at 60 min of the reaction, and then remained constant until the end of the experiment. A similar behavior was observed for GLA (15.7 mol.%), while AA and OA suffered a slight increase with the course of the reaction. Although GA is produced from the beginning of the reaction, with high selectivity, it is over-oxidized by C–C splitting mainly due to the presence of peroxide radicals in the reaction medium. This is in accordance with results previously reported by Kumar et al. [8].

In general, these results indicate that the reaction does not evolve properly in the absence of a solid catalyst in the reaction medium, indicating that the presence of the catalyst is essential for the development of GLY oxidation.

## 4. Conclusions

Noble metals modified ZSM-11 zeolites were successfully used for the conversion of glycerol towards higher value-added products. Pt-ZSM-11 achieved the highest GLY conversion (≈67 mol.%), followed by the Au-ZSM-11 (≈50 mol.%) and Pd-ZSM-11 (38 mol.%) catalysts. The exhibited behavior is due to a higher Brønsted/Lewis relationship observed in the catalysts as well as a small particle size; therefore, a better dispersion of the metallic species would favor the catalytic performance of the catalysts.

The Pt-ZSM-11 catalyst was selective in the oxidation of GLY toward LA (≈30 mol.%), while the Pd-ZSM-11 and Au-ZSM-11catalysts resulted in higher selectivity to GA (>70 mol.% and >50 mol.%, respectively).

The conversion observed when using bimetallic systems was lower than that obtained with monometallic catalysts; the Au-Pt ZSM-11 catalyst achieved ≈45 mol.% selectivity to LA, higher than that achieved with the Pt-ZSM-11 sample. The presence of both Au and Pt in the same catalyst leads to a combined effect, directing the reaction towards obtaining LA, via DHA.

The main reaction parameters were studied by using Pt-ZSM-11 in order to obtain the optimal reaction conditions for accurate maximum GLY conversion and LA selectivity. Thus, optimal operating conditions, such as temperature of 100 °C, NaOH/GLY molar ratio of 2, and reaction time of 240 min allowed attaining a high yield to the desired product (LA) with low formation of over-oxidation products.

Finally, it can be concluded that in the absence of a solid catalyst, the reaction does not evolve into the desired products, indicating that the metal active site adequately isolated in the zeolite matrix is essential in the glycerol oxidation mechanism presented above.

## Figures and Tables

**Figure 1 nanomaterials-11-00510-f001:**
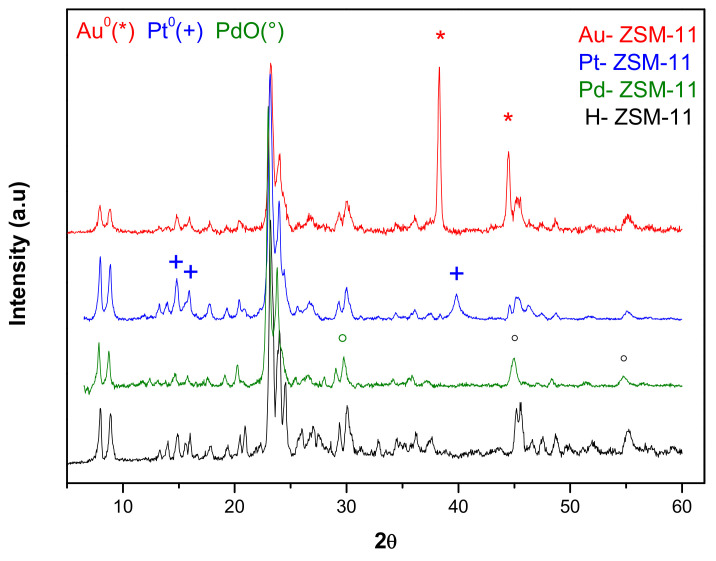
X-ray diffraction (XRD) patterns of monometallic catalysts Au-ZSM-11, Pt-ZSM-11, and Pd-ZSM-11, compared to the pristine H-ZSM-11 matrix.

**Figure 2 nanomaterials-11-00510-f002:**
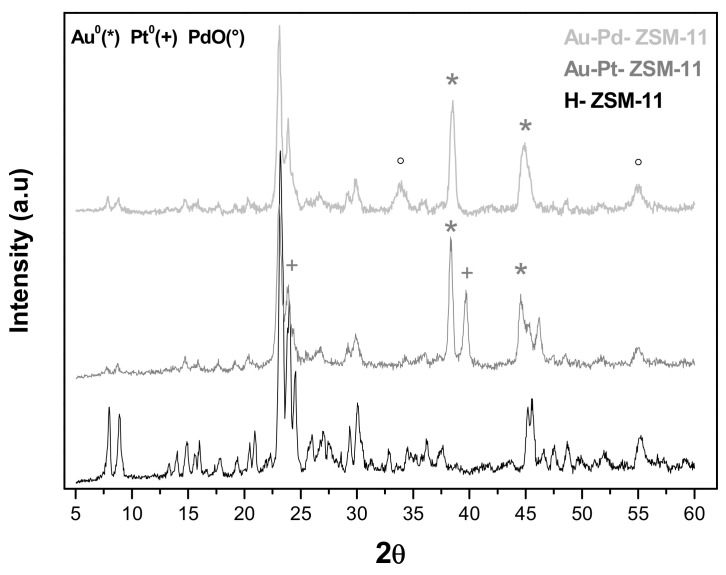
X-ray diffraction (XRD) patterns of bimetallic catalysts Au-Pt-ZSM-11 and Au-Pd-ZSM-11, compared to the pristine H-ZSM-11 matrix.

**Figure 3 nanomaterials-11-00510-f003:**
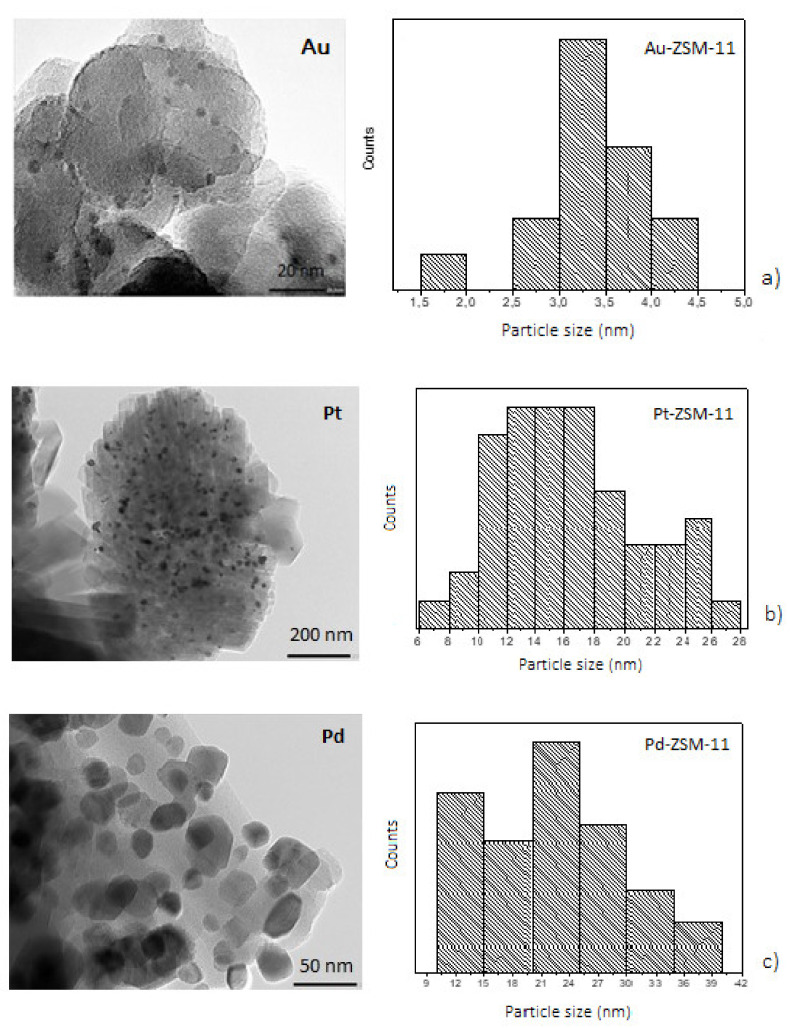
Transmission electron microscopy (TEM) images and particle size distribution of the monometallic catalysts; (**a**) Au- ZSM-11, (**b**) Pt-ZSM-11, (**c**) Pd-ZSM-11.

**Figure 4 nanomaterials-11-00510-f004:**
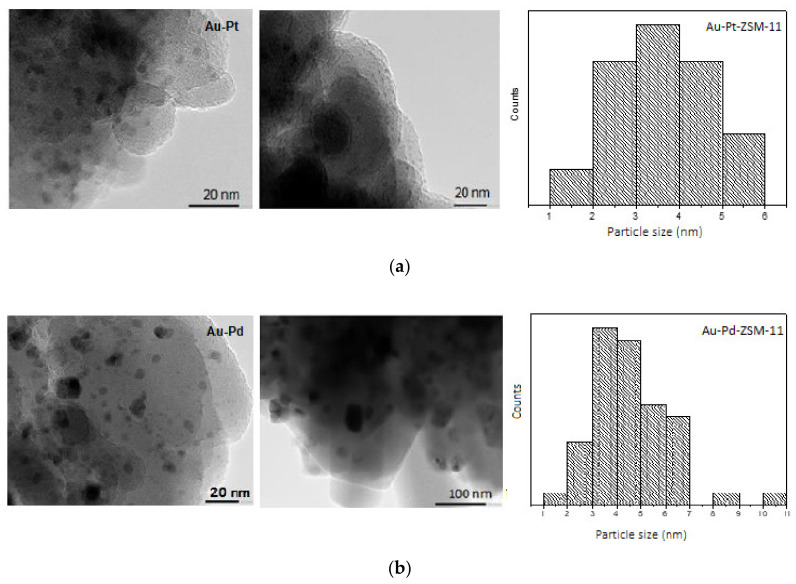
Transmission electron microscopy (TEM) images and particle size distribution of the bimetallic catalysts; (**a**) Au-Pt-ZSM-11, and (**b**) Au-Pd-ZSM-11.

**Figure 5 nanomaterials-11-00510-f005:**
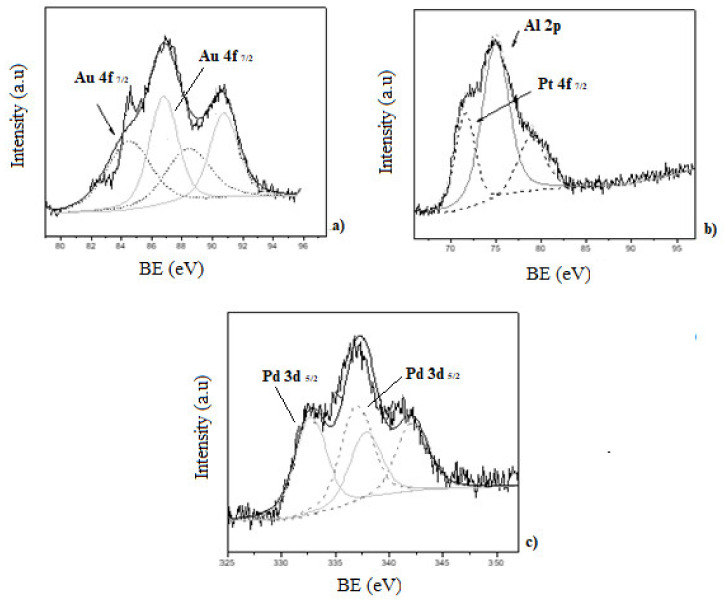
X-ray photoelectron spectroscopy (XPS) spectra of the monometallic catalysts; (**a**) Au-ZSM-11; (**b**) Pt-ZSM-11; (**c**) Pd-ZSM-11.

**Figure 6 nanomaterials-11-00510-f006:**
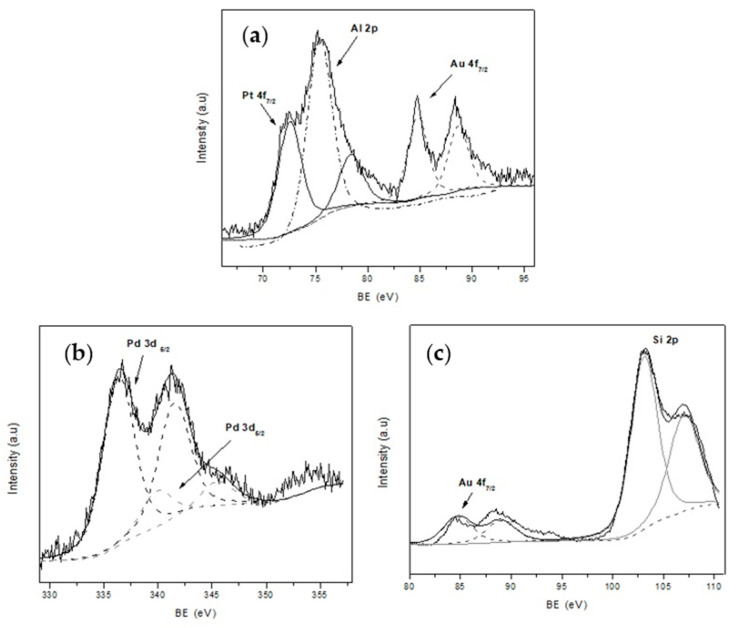
X-ray photoelectron spectroscopy (XPS) spectra of bimetallic catalysts; (**a**) Au-Pt- ZSM-11; (**b**) and (**c**) Au-Pd-ZSM-11.

**Figure 7 nanomaterials-11-00510-f007:**
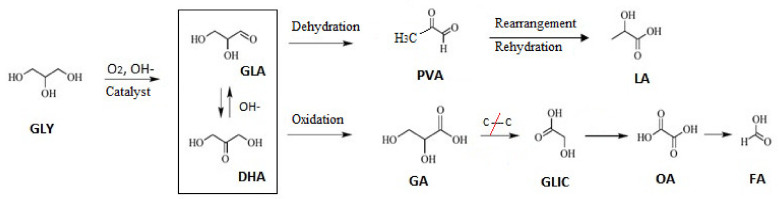
Scheme of reaction pathways for GLY oxidation.

**Figure 8 nanomaterials-11-00510-f008:**
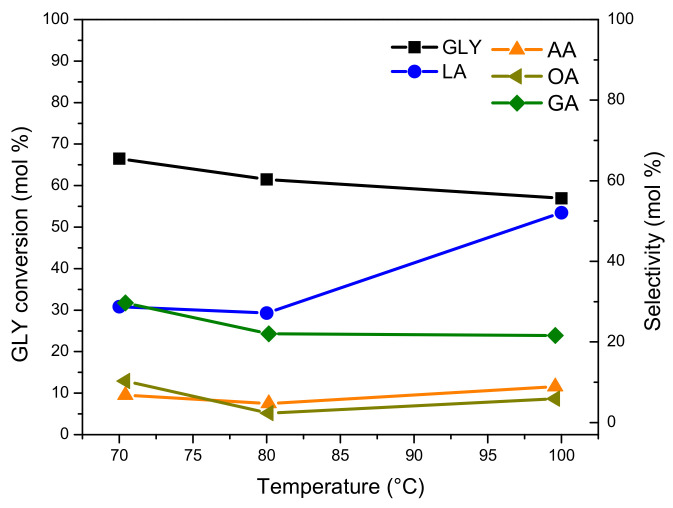
GLY conversion and oxidation products’ selectivity vs. reaction temperature in GLY oxidation of the Pt-ZSM-11 catalyst.

**Figure 9 nanomaterials-11-00510-f009:**
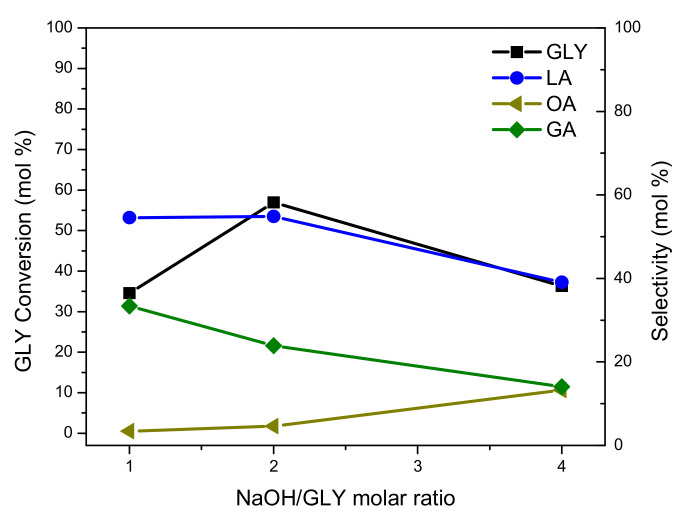
GLY conversion and oxidation products selectivity vs. NaOH/GLY molar ratio in GLY oxidation of the Pt-ZSM-11 catalyst.

**Figure 10 nanomaterials-11-00510-f010:**
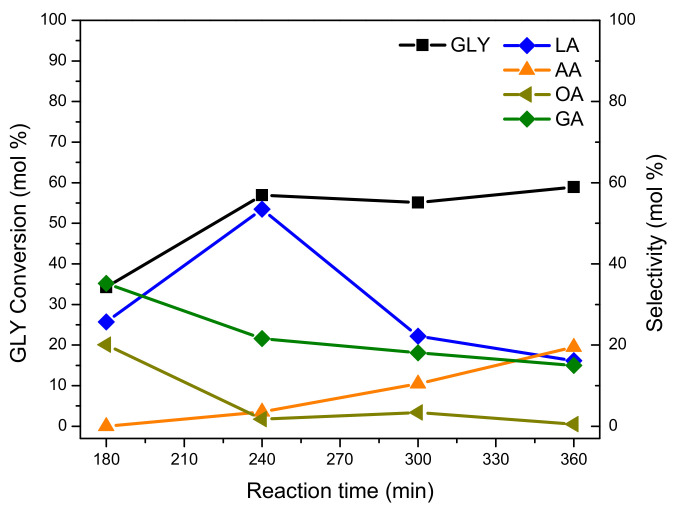
GLY conversion and oxidation products’ selectivity vs. reaction time in GLY oxidation of the Pt-ZSM-11 catalyst.

**Figure 11 nanomaterials-11-00510-f011:**
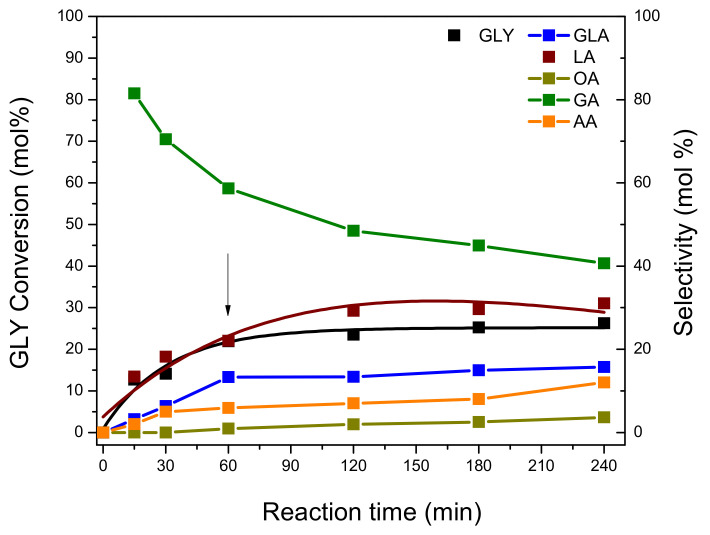
GLY conversion and selectivity to products vs. reaction time with Pt-ZSM-11 catalyst removal from the reaction.

**Table 1 nanomaterials-11-00510-t001:** The main physicochemical and textural properties of ZSM-11 zeolite impregnated with noble metals.

Catalyst	Surface Area (m^2^/g)	Metal Loading (wt.%)		Acid Sites (μmol of Site/g Catalyst)
		ICP		FTIR
	BET	M_1_	M_2_	Brønsted/Lewis
H-ZSM-11	392	-	-	9.26
Pt-ZSM-11	383	1.18 *	-	4.76
Pd-ZSM-11	370	0.77 *	-	3.95
Au-ZSM-11	392	0.66	-	4.12
Au-Pt-ZSM-11	378	1.21	0.98	3.36
Au-Pd-ZSM-11	382	1.13	1.05	2.19

M_1_: impregnated metal (Pt, Pd, Au); M_2_: Au incorporated by co-impregnation; Values determined by ICP, then corroborated by X-ray Fluorescence (XRF).

**Table 2 nanomaterials-11-00510-t002:** Surface metal loading of monometallic and bimetallic catalysts.

	Superficial Metal Loading (% Atomic) XPS					
	Au 4f_7/2_ (eV)		Pt **4f_7/2_ (eV)**		Pd 3d_5/2_ (eV)	
Catalysts	Au^0^ (84.4)	Au^+1^ (86.7)	Pt^0^ (71.5)	Pt^+2^ (72.6)	Pd^0^ (336.98)	Pd^+2^ (337.98)
Pt-ZSM-11	-	-	0.1	-	-	-
Pd-ZSM-11	-	-	-	-	0.1	0.06
Au-ZSM-11	0.1	0.22	-	-	-	-
Au-Pt-ZSM-11	0.17	-	-	0.33	-	-
Au-Pd-ZSM-11	0.43	-	-	-	0.83	0.1

**Table 3 nanomaterials-11-00510-t003:** Performance of different metal-supported catalysts in GLY oxidation after 4 h of reaction.

		Products Selectivity (mol.%)						
Catalyst	GLY Conversion (mol.%)	GLA	GLIC	LA	GA	AA	OA	NI
Pt-ZSM-11	66.5	20.8	-	30.4	29.7	6.8	10.2	1.9
Au-ZSM-11	50.5	24.4	23.9	-	51.7	-	-	-
Pd-ZSM-11	38.0	1.2	0.8	-	74.0	-	-	23.9
Au-Pt-ZSM-11	35.3	21.5	-	45.1	30.5	1.2	1.5	
Au-Pd-ZSM-11	27.4	21.6	-	26.9	31.2	0.4	17.6	2.2

Reaction conditions: GLY solution (0.25 M), temperature of 70 °C, NaOH/GLY molar ratio = 2, O_2_ flow 100 mL/min, 200 mg of catalyst, reaction time of 4 h.

## Data Availability

Not applicable.

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
