# Peer review of "Conversion of Glycerol to Value Added Products in a Semi-Continuous Batch Reactor Using Noble Metals Supported on ZSM-11 Zeolite"

_nanomaterials, 2021, doi:10.3390/nano11020510_

Round 1

Reviewer 1 Report

In this manuscript, Diguilio and coworkers report the preparation of Au, Pt, Pt, Au/Pt, and Au/Pd-loaded ZSM-11 catalysts and their catalytic performance in the oxidation of glycerol (GLY) to lactic acid (LA). The selectivity and activity of the catalytic systems were investigated and Pt-ZSM-11 showed the relatively good activities and selectivities. My comments are as follows.

  1. In ref. 18, this research group has published the closely-related paper on the Cu and Cr-ZSM-11 for the GLY oxidation at mild conditions. As described in the introduction part, other scientists have reported the metal-incorporated catalysts for the oxidation of GLY. Unfortunately, the authors did not describe strictly the merits of noble metal-ZSM-11 catalysts, compared to Cu and Cr-ZSM-11 and noble metal catalysts reported in the literature. Thus, in reading this manuscript, it was hard to figure out the academic intention of authors about the novel points. For me, the contents of this manuscript are technical reports on the new entries of catalysts. Usually, the scientists may think that the development of the non-noble metal catalysts is more meaningful than the noble metal-based ones.Unfortunately, considering the present situation, I cannot find merits and academic meaning of the noble metal-ZSM-11 catalysts for the GLY oxidation. I suggest that the authors should focus on novel academic points. For example, the comparison of noble metal-ZSM-11 vs Cu-, Cr-ZSM-11 catalysts or the comparison of noble metal-ZSM-11 catalysts vs noble metal on the solid supports in the literature can be one of the subjects. After focusing and clarifying such subjects, if the new scientific principles can be disclosed, the work can be re-submitted. In addition, if the authors want to report the ZSM-11 as a new solid support, compared to the reported solid supports (such as carbons and etc), they should supply the comparison studies with the careful optimization of the kinds and amounts of metals. Based on the results, the support effect can be discussed.
  2. Please add the performance comparison table of the catalysts in the literature to the SI.
  3. Based on the comparison of the catalytic performance, please elucidate and describe the possible new scientific principles on the advanced performances of the catalysts of the manuscript.
  4. The catalysts preparation and catalytic tests do not provide the experimental details. To be a scientific document and to help the easy reproduction of scientists, all g and mL values used in the catalyst synthesis and catalytic reactions should be carefully supplied in the experimental sections. Without these information, the meaning of the claims and discussion on the text can be limited.

Author Response

Point 1: In ref. 18, this research group has published the closely-related paper on the Cu and Cr-ZSM-11 for the GLY oxidation at mild conditions. As described in the introduction part, other scientists have reported the metal-incorporated catalysts for the oxidation of GLY. Unfortunately, the authors did not describe strictly the merits of noble metal-ZSM-11 catalysts, compared to Cu and Cr-ZSM-11 and noble metal catalysts reported in the literature. Thus, in reading this manuscript, it was hard to figure out the academic intention of authors about the novel points. For me, the contents of this manuscript are technical reports on the new entries of catalysts. Usually, the scientists may think that the development of the non-noble metal catalysts is more meaningful than the noble metal-based ones.Unfortunately, considering the present situation, I cannot find merits and academic meaning of the noble metal-ZSM-11 catalysts for the GLY oxidation. I suggest that the authors should focus on novel academic points. For example, the comparison of noble metal-ZSM-11 vs Cu-, Cr-ZSM-11 catalysts or the comparison of noble metal-ZSM-11 catalysts vs noble metal on the solid supports in the literature can be one of the subjects. After focusing and clarifying such subjects, if the new scientific principles can be disclosed, the work can be re-submitted. In addition, if the authors want to report the ZSM-11 as a new solid support, compared to the reported solid supports (such as carbons and etc), they should supply the comparison studies with the careful optimization of the kinds and amounts of metals. Based on the results, the support effect can be discussed.

Response to point 1.

Thank you very much for your valuable suggestions that have allowed us to substantially improve the manuscript. We have changed the text and added a new paragraph (lines 52-64 in the revised version) in which performance of other reaction systems, also including different catalytic supports are mentioned. It is worth noting that, although some of them exhibit better performance than the catalysts reported in our work, the operating conditions are very different from ours, such as the use of more sophisticated reactors or more drastic reaction conditions (i.e. higher temperatures).

As the reviewer has also suggested, we have described in more detail the results obtained when using transition metals such as Cu and Cr on different zeolitic supports (old reference 18, now reference 19 in the revised version), thus allowing analyzing the merits of using noble metals, as well as the zeolitic support ZSM-11. This information has been added in lines 75-77 of the revised version of the manuscript.

On the other hand, as suggested by the reviewer, a paragraph was added (lines 81-90 in the revised version) explaining the advantage of using ZSM-11 zeolites as catalytic support.

Point 2: Please add the performance comparison table of the catalysts in the literature to the SI.

Response to point 2.

Many thanks for the suggestion. The corresponding comparative Table (S4) was added in the revised SI document.

Point 3: Based on the comparison of the catalytic performance, please elucidate and describe the possible new scientific principles on the advanced performances of the catalysts of the manuscript.

Response to point 3.

Following the Reviewer suggestion, the corresponding explanatory text was added at the end of the Introduction section (lines 111-114 in the revised version).

Point 4: The catalysts preparation and catalytic tests do not provide the experimental details. To be a scientific document and to help the easy reproduction of scientists, all g and mL values used in the catalyst synthesis and catalytic reactions should be carefully supplied in the experimental sections. Without these information, the meaning of the claims and discussion on the text can be limited.

Response to point 4.

More details were added in the experimental description of the synthesis of catalysts. This information has been reported in the experimental section lines 117-119 of the revised version of the manuscript.

Reviewer 2 Report

The manuscript deals with the preparation, characterization and testing of noble metals catalysts for the oxidation of glycerol to valuable products. The paper suffers from lacks avoiding its publication and it must be modified according to the comments reported below.

  • Abstract, line 18. Au is reported twice.
  • mass balance is never reported.
  • Figure 2 is wrong. It is equal to figure 1
  • Section 3.2. Catalytic activities are compared at different GLY conversions. Accordingly, selectivities cannot be adequately compared, especially if series reactions occurs (as in the current case).
  • Section 3.4.1. Why did the Authors excluded mass transfer limitations to explain the effect of the temperature?
  • In the reaction mechanism overoxidation products must be included, as evidenced by the effect of the reaction time.
  • Section 3.5 is not indicative of the catalyst stability. It is related to the occurrence of homogeneous reactions.
  • The manuscript must be carefully proofread. There are several typos and errors.

Reviewer 3 Report

Manuscript Number: nanomaterials-1077633

Title: Conversion of glycerol to value added products in a semi-continuous batch reactor using noble metals supported on ZSM-11 zeolite

The scope of this work was to investigate Au, Pt and Pd supported on ZSM-11 microporous zeolite as catalysts for glycerol (GLY) oxidation towards higher value-added products. To this end, ZSM-11 was synthesized by hydrothermal treatment, whereas ion exchange with NH4Cl was performed to recover acidic sites and then, Au, Pt and Pd were incorporated onto this material by wet impregnation procedure. The materials were characterized by different techniques, such as XRD, ICP, TEM- XEDS, and XPS, and they were evaluated in the glycerol oxidation reaction by using alkaline medium and molecular oxygen as oxidizing agent. It has been proven that the higher conversion of GLY (66.5 mol.%) was reached over the Pt–ZSM-11 catalyst with moderate selectivity towards lactic acid (LA), while the bimetallic Au-Pt-ZSM-11 catalyst offered high selectivity to LA at moderate GLY conversion. Furthermore, optimization of main reaction parameters (i.e. temperature, reaction time and NaOH/GLY ratio) was carried out to maximize the selectivity towards the LA desired product. The article is quite interesting; it includes enough work and adds some new information especially on the conversion of glycerol to value added products in a semi-continuous batch reactor using noble metals supported on ZSM-11 zeolite. Moreover, it adheres to the journal’s standards as the scope of nanomaterials covers the preparation, characterization and application of all nanomaterials. Thus, my recommendation is to be accepted after major revision.

Specific Comments

- English should be improved by a native speaker

- Authors should explain in details the novelty of this work. Furthermore, a more in-depth discussion on the reasons of the differences on catalysts’ performance is needed

- More recent works from the literature should be added and discussed through this manuscript. Indicatively papers:

  1. A.C. Dimian, C.S. Bildea, Sustainable process design for manufacturing acrylic acid from glycerol, Chemical Engineering Research and Design 1 6 6 ( 2 0 2 1 ) 121–134
  2. Charisiou N.D., Italiano C., Pino L., Sebastian V., Vita A., Goula M.A., Hydrogen production via steam reforming of glycerol over Rh/γ-Al2O3 catalysts modified with CeO2, MgO or La2O3. Renewable Energy, 2020, 162, pp. 908–925
  3. M. El Doukkali, A. Iriondo, I. Gandarias, Review - Enhanced catalytic upgrading of glycerol into high value-added H2 and propanediols: Recent developments and future perspectives, Molecular Catalysis 490 (2020) 110928
  4. Charisiou N.D., Siakavelas G.I., Papageridis K.N., Motta D., Dimitratos N., Sebastian V., Polychronopoulou K., Goula M.A., The effect of noble metal (M: Ir, Pt, Pd) on M/Ce2O3-γ-Al2O3 catalysts for hydrogen production via the steam reforming of glycerol. Catalysts 10 (2020) 790
  5. R.J. Chimentao, P. Hirunsit, C.S. Torres, M. Borges Ordono, A. Urakawa, J.L. G. Fierro, D. Ruiz, Selective dehydration of glycerol on copper based catalysts, Catalysis Today xxx (xxxx) xxx
  6. Maria Cristina de Almeida Silva, Luiz Olinto Monteggia, Jose Carlos Alves Barroso Júnior, Camille Eichelberger Granada, Adriana Giongo, Evaluation of semi-continuous operation to hydrogen and volatile fatty acids production using raw glycerol as substrate, Renewable Energy 153 (2020) 701e710
  7. Lais F. Oton, Alcineia C. Oliveira, Samuel Tehuacanero-Cuapa, Gilberto D. Saraiva, Francisco F. de Sousa, Adriana Campos, Gian Duarte, Joao R. Bezerra, Catalytic acetalization of glycerol to biofuel additives over NiO and Co3O4 supported oxide catalysts: experimental results and theoretical calculations, Molecular Catalysis 496 (2020) 111186
  8. Andrii Kostyniuk, David Bajec, Petar Djinović, Blaž Likozar, One-step synthesis of glycidol from glycerol in a gas-phase packed-bed continuous flow reactor over HZSM-5 zeolite catalysts modified by CsNO3, Chemical Engineering Journal 394 (2020) 124945
  9. Polychronopoulou K., Charisiou N.D., Siakavelas G., AlKhoori A.A., Sebastian V., Hinder S.J., Baker M.A., Goula M.A., Ce-Sm-xCu cost efficient catalysts for H2 production through the glycerol steam reforming reaction. Sustainable Energy & Fuels 3 (2019) 673-691.
  10. Jiaxiong Liu, Yajin Li, Huimin Liu, Dehua He, Transformation of CO2 and glycerol to glycerol carbonate over CeO2-ZrO2 solid solution - effect of Zr doping, Biomass and Bioenergy 118 (2018) 74–83
  11. Charisiou N.D., Polychronopoulou K., Asif A., Goula M.A., The potential of glycerol and phenol towards H2 production using steam reforming reaction: A review. Surface and Coatings Technology 352 (2018) 92-111
  12. A. Lähde, R.J. Chimentão, T. Karhunen, M.G. Álvarez, J. Llorca, F. Medina, J. Jokiniemi, L.B. Modesto-López, Co-Al spinel-based nanoparticles synthesized by flame spray pyrolysis for glycerol conversion, Advanced Powder Technology 28 (2017) 3296–3306

Author Response

Point 1: English should be improved by a native speaker.

Response to point 1.

Some improvements in the English version were done in the revised version of the manuscript.

Point 2: Authors should explain in details the novelty of this work. Furthermore, a more in-depth discussion on the reasons of the differences on catalysts’ performance is needed.

Response to point 2.

Following your valuable suggestions (and these form Reviewer #1), we have changed the text and added a new paragraph (lines 52-64 in the revised version) in which performance of other reaction systems, also including different catalytic supports are mentioned. It is worth noting that, although some of them exhibit better performance than the catalysts reported in our work, the operating conditions are very different from ours, such as the use of more sophisticated reactors or more drastic reaction conditions (i.e. higher temperatures). In addition, we have described in more detail the results obtained when using transition metals such as Cu and Cr on different zeolitic supports (old reference 18, now reference 19 in the revised version), thus allowing analyzing the merits of using noble metals, as well as the zeolitic support ZSM-11. This information has been added in lines 75-77 of the revised version of the manuscript. Finally, the corresponding text pointing out the relevance of the here presented work was added at the end of the Introduction section (lines 111-114 in the revised version).

Point 3: More recent works from the literature should be added and discussed through this manuscript.

Response to point 3.

Thank you for the list of very interesting and recent literature related to the valorization of glycerol to different products, such as acrylic acid, hydrogen, glycerol carbonate, dehydrated and acetylated derivatives, among others. Nevertheless, any of the twelve papers here listed related to glycerol oxidation process, which is the process discussed in our research work. At this point, we consider that the citation of the published articles provided by the Reviewer is not necessary in this case.

Reviewer 4 Report

In this work, the authors studied nanosized noble metal catalyst (Au, Pt, and Pd) supported on zeolite for glycerol oxidation towards higher value-added products. Monometallic (Au, Pt, and Pd) or bimetallic catalyst (Au-Pt, and Au-Pd) on microporous zeolite was synthesized by wet impregnation method. Furthermore, optimization of the main reaction parameters was searched to maximize the selectivity towards the Lactic acid. These catalysts showed the enhanced conversion of glycerol with higher value-desired products. However, this paper failed to show any originality by just using well-known catalysts and supports. Unless the authors provide any new insights or development in the manuscript, it should be reconsidered for publication.

Lastly, this reviewer thinks that this manuscript needs to prove some questions followed below.

    1. The synthesized catalysts were characterized by different techniques for studying their physical and chemical properties. For showing the alloying of metal catalysts, the authors presented XRD patterns in Figure 1 and Figure 2. Please check the Figure 1 and Figure 2. It is only changed the index of the figure, but the data is not changed.
    2. The authors investigated the formation of alloys of Au-Pt and Au-Pd by XEDS analysis. However, it is hard to figure out that it proves the alloy of the bimetallic catalyst due to less-matched spots. Could you show the other method to prove the formation of an alloy of bimetallic catalyst?
    3. In a catalytic test, the bimetallic catalyst presented lower conversion than that obtained with monometallic catalyst. The only selectivity can be improved using the bimetallic catalyst. However, conversion is an important factor in the catalytic system. Could you give a reason for the lower conversion rate?
    4. After the catalytic test, the catalyst does not go through the aggregation or degradation of particles?   
    5. Please enhance the resolution of graphs in Figure 3, and Figure 4.
    6. Please check the following typo in the manuscript.
  • Please change Au-, Pt-. and Au- ZSM-11 to Au, Pt-, and Pd-ZSM-11 in Abstract pages (18th line).
  • Please check reference parts (No doi number): 3, 5, 6, 7, 8, (Check the title): 21
  • (No journal name): 4, 5, 6, 7, 8, 17, 29, 31, 32, 37, 39
  • (No bold style): 2, 23, 25

Author Response

Point 1: The synthesized catalysts were characterized by different techniques for studying their physical and chemical properties. For showing the alloying of metal catalysts, the authors presented XRD patterns in Figure 1 and Figure 2. Please check the Figure 1 and Figure 2. It is only changed the index of the figure, but the data is not changed.

Response to point 1.

The comment made by the reviewer is correct. Figure 2 has been replaced by the correct Figure. We apologize for the mistake made.

Figure 2. XRD patterns of bimetallic catalysts Au-Pt-ZSM-11 and Au-Pd-ZSM-11 compared with pristine H-ZSM-11 matrix.

Point 2: The authors investigated the formation of alloys of Au-Pt and Au-Pd by XEDS analysis. However, it is hard to figure out that it proves the alloy of the bimetallic catalyst due to less-matched spots. Could you show the other method to prove the formation of an alloy of bimetallic catalyst?

Response to point 2.

TEM images of both catalysts showed darker wide areas that would be indicative the interaction between two metals (Au-Pt and Au-Pd). By XEDS analysis of a representative area on catalytic surface, the closeness of metallic species Au, Pt and Pd could be observed, especially in bimetallic catalysts. This could indicate the formation of alloys of Au-Pt and Au-Pd. A more detailed study should be performed to confirm the presence of such an alloy by lattice spacing calculation. That could be quantifying the lattice spacing using aberration-corrected atomic resolution STEM images of the bimetallic catalysts as reported in reference [8]. On the other hand, the decrease in metal particles size in bimetallic catalysts is also indicative of a closeness between Au-Pt and Au-Pd, being able to form an alloy or bimetallic particles, as reported in reference [29].

Anyway we consider that it is important to continue working on the synthesis of the catalysts as well as on their subsequent characterization techniques.

Point 3: In a catalytic test, the bimetallic catalyst presented lower conversion than that obtained with monometallic catalyst. The only selectivity can be improved using the bimetallic catalyst. However, conversion is an important factor in the catalytic system. Could you give a reason for the lower conversion rate?

Response to point 3.

Although the selectivity towards LA has increased when using the Au-Pt-ZSM-11 and Au-Pd-ZSM-11 bimetallic catalysts, it is true that the conversion observed is lower than that obtained using the monometallic samples. This behavior could be related to the lower relationship of Brønsted/Lewis acid sites present in the bimetallic samples; these values being between 29% in the Au-Pt-ZSM-11 zeolite and 44% in the Au-Pd-ZSM-11 zeolite lower than these encountered for the Pt-ZSM-11 and Pd-ZSM-11, respectively. An explanatory text was added in the revised version of the manuscript (see lines 366-369).

Point 4: After the catalytic test, the catalyst does not go through the aggregation or degradation of particles?

Response to point 4.

In this work we have determined that the catalyst do not suffer leaching of Pt species during the process, thus demonstrating the stability and resistance of the metal-zeolite catalyst under the reaction conditions employed. It is probably that Pt nanoparticles could aggregate in some extent during reaction in the zeolitic matrix. Unfortunately, we have not performed surface analysis of the used catalyst in this work. We understand that it is important to analyze the solid catalyst after reaction to evaluate whether it is feasible to reuse it without prior treatment.

Point 5: Please enhance the resolution of graphs in Figure 3, and Figure 4.

Response to point 5.

The resolution of the graphs in Figures 3 and 4 have been improved.

Point 6: Please check the following typo in the manuscript.

Response to point 6.

We apologize for the typo in the manuscript, they have already been corrected. In addition, the changes suggested by the Reviewer have been made in the indicated bibliographic references.

Round 2

Reviewer 1 Report

Although the authors used the noble metals in this work, the catalytic performance is not impressive, compared to the previous noble metal-free reports of authors. Also, the comparison table did not supply the information of experimental conditions.

I cannot find the suggestion of authors for the novel scientific principles.

The authors did not supply the performance comparison of the catalytic sytems in the literature (of other scientists). Thus, I could not judge the significance of this work.

The authors did not supply the experimental g and mL values of all reagents used in the work. Please note that the relative values such as ratios and compositions make trial and errors in the reporduction of readers. Please provide all g and mL values for readers.

Author Response

Answers to comments of Reviewer

 Reviewer comment:

Although the authors used the noble metals in this work, the catalytic performance is not

impressive, compared to the previous noble metal-free reports of authors. Also, the comparison table did not supply the information of experimental conditions.

Answer:

Thank you very much for your valuable suggestions. We have made the following changes in the new revised version of manuscript and in the SI to afford the concerns of the Reviewer:

  • In order to emphasize the significance of our research work, mainly related to the use of noble metal supported on zeolites type catalysts under very mild reaction conditions (low temperature and pressure, use of molecular O2), the following texts were added in the Introduction section:

In lines 81-83: “In this sense, the use of less expensive oxidants, such as molecular O2 or air, as well as the employment of mild reaction conditions (low temperature and pressure) are highly preferred.”

In lines 115 to 118: “… towards lactic acid (LA). Particularly, reactions are performed under very mild reaction conditions (at low temperature and atmospheric pressure) and by using molecular O2 as oxidant, thus offering the possibility to obtain an interesting product, such as LA at industrially applicable conditions.”

  • The Table in section S4 of the Supporting Information has been extended by including data obtained at 1 and 2 hours of reactions for both series of catalysts noble metals (Pt, Pd and Au) and Cu and Cr supported zeolites. The corresponding reaction conditions have been added in all the cases.

  • In addition, the following text has been added in the discussion of catalytic results (section 3.2, lines 386 to 397) of the new revised manuscript:

“Comparison of the results obtained with Pt/ZSM-11 and Au-Pt/ZSM-11 with those previously reported by us with transition metals (Cr, Cu) supported zeolites, it can be said that with both reaction systems, similar yields were achieved to the desired products (see tables of S4 section in SI). Particularly, LA yields between 15-20% at 1-2 hours of reactions are achieved with both noble metals and transition metals supported on ZSM-11 zeolite. Nevertheless, it is important to remark that when using noble metals supported on ZSM-11 milder reaction conditions have been selected (atmospheric pressure and low temperatures) compared to those noble metals supported systems previously reported by other authors [36-41]. It is clear that higher reaction pressures and temperatures have an effect on the final GLY conversion and the selectivity of interest products, thus overcoming the susceptibility of noble metals to poisoning and leaching in the reaction medium (see section S5 in SI). However, the results reached in this work were promising and offer potential to continue finding optimal conditions to obtain higher yields.”

  • The corresponding references [36 to 41] have been added in the new revised version of the manuscript, and the old references re-numbered accordingly.

  • Finally, a new section S5 has been included in the SI.

Reviewer comment:

I cannot find the suggestion of authors for the novel scientific principles.

Answer:

Many thanks for the remark. The following text has been added in section 3.3 (lines 435 to 445):

“With all these data and observations in mind, it is possible to stress that the presence of highly and homogeneously dispersed noble metals nanoparticles in our Pt/ZSM-11 and Au-Pt/ZSM-11 catalysts allow producing the dehydrogenation reaction of GLY to GLA/DHA helped by the OH- species given by the basic medium. Then, the equilibrium GLA/DHA is shifted towards the GLA for the later production of LA via dehydration, mainly carried out onto the Brönsted/Lewis acid sites of the ZSM-11 zeolite. It is true that once the carbonyl (aldehyde or ketone) intermediate is formed on catalyst surface, it would undergo the elimination of β-hydride to generate the corresponding carboxylic acid, which is also facilitated by the existing surface bound hydroxide formed during the first dehydrogenation step. Nevertheless, the use of very mild reaction conditions reduce in some extent the production of other oxidation products, also avoiding the C-C rupture occurring due to over-oxidation reactions that derives in the formation of C1-C2 carboxylic acids.”

Reviewer comment:

The authors did not supply the performance comparison of the catalytic systems in the literature (of other scientists). Thus, I could not judge the significance of this work.

Answer:

Following the Reviewer suggestion, a new explanatory text was added in the discussion of catalytic results (section 3.2, lines 386 to 397) of the new revised manuscript (see also answer to first Reviewer’s comment):

“Comparison of the results obtained with Pt/ZSM-11 and Au-Pt/ZSM-11 with those previously reported by us with transition metals (Cr, Cu) supported zeolites, it can be said that with both reaction systems, similar yields were achieved to the desired products (see tables of S4 section in SI). Particularly, LA yields between 15-20% at 1-2 hours of reactions are achieved with both noble metals and transition metals supported on ZSM-11 zeolite. Nevertheless, it is important to remark that when using noble metals supported on ZSM-11 milder reaction conditions have been selected (atmospheric pressure and low temperatures) compared to those noble metals supported systems previously reported by other authors [36-41]. It is clear that higher reaction pressures and temperatures have an effect on the final GLY conversion and the selectivity of interest products, thus overcoming the susceptibility of noble metals to poisoning and leaching in the reaction medium (see section S5 in SI). However, the results reached in this work were promising and offer potential to continue finding optimal conditions to obtain higher yields.”

Reviewer comment:

The authors did not supply the experimental g and m L values of all reagents used in the work. Please note that the relative values such as ratios and compositions make trial and errors in the reproduction of readers. Please provide all g and ml values for readers.

Answer:

More details (including g and mL of reactants used) have been added in the experimental description of the catalytic tests (see lines 176-177 and 190 of the new revised version of the manuscript).

Reviewer 3 Report

I am satisfied that the authors carried out the changes requested

Author Response

Thank you for your valuable suggestions and remarks.

Reviewer 4 Report

Well revised. The paper is ready to be published.

Author Response

(The authors gave the same response as above.)
